# Photosynthetic Responses of Peat Moss (*Sphagnum* spp.) and Bog Cranberry (*Vaccinium oxycoccos* L.) to Spring Warming

**DOI:** 10.3390/plants13223246

**Published:** 2024-11-19

**Authors:** Michal Antala, Abdallah Yussuf Ali Abdelmajeed, Marcin Stróżecki, Włodzimierz Krzesiński, Radosław Juszczak, Anshu Rastogi

**Affiliations:** 1Laboratory of Bioclimatology, Department of Ecology and Environmental Protection, Poznan University of Life Sciences, 60-649 Poznan, Poland; abdallah.abdelmajeed@up.poznan.pl (A.Y.A.A.); marcin.strozecki@up.poznan.pl (M.S.); radoslaw.juszczak@up.poznan.pl (R.J.); 2Department of Vegetable Crops, Poznan University of Life Sciences, 60-995 Poznan, Poland; wlodzimierz.krzesinski@up.poznan.pl

**Keywords:** bog cranberry, peat moss, climate change, peatland, photosynthesis

## Abstract

The rising global temperature makes understanding the impact of warming on plant physiology in critical ecosystems essential, as changes in plant physiology can either help mitigate or intensify climate change. The northern peatlands belong to the most important parts of the global carbon cycle. Therefore, knowledge of the ongoing and future climate change impacts on peatland vegetation photosynthesis is crucial for further refinement of peatland or global carbon cycle and vegetation models. As peat moss (*Sphagnum* spp.) and bog cranberry (*Vaccinium oxycoccos* L.) represent some of the most common plant functional groups of peatland vegetation, we examined the impact of experimental warming on the status of their photosynthetic apparatus during the early vegetation season. We also studied the differences in the winter to early spring transition of peat moss and bog cranberry photosynthetic activity. We have shown that peat moss starts photosynthetic activity earlier because it relies on light-dependent energy dissipation through the winter. However, bog cranberry needs a period of warmer temperature to reach full activity due to the sustained, non-regulated, heat dissipation during winter, as suggested by the doubling of photosystem II efficiency and 36% decrease in sustained heat dissipation between the mid-March and beginning of May. The experimental warming further enhanced the performance of photosystem II, indicated by a significant increase in the photosystem II performance index on an absorption basis due to warming. Therefore, our results suggest that bog cranberry can benefit more from early spring warming, as its activity is sped up more compared to peat moss. This will probably result in faster shrub encroachment of the peatlands in the warmer future. The vegetation and carbon models should take into account the results of this research to predict the peatland functions under changing climate conditions.

## 1. Introduction

Due to anthropogenic activities, the climate is changing faster than humankind could ever experience. The temperatures in the recent decades continue to rise at an unprecedented pace [1,2]. The period from June 2023 to May 2024 was the warmest year ever recorded for the globe, as May was already the 12th month in a row that was the warmest month from the beginning of records [3]. Because the climate projections show a further increase in the temperature over the next few years, it is crucial to understand the impact of the warming on the plants’ physiology in critical ecosystems that may mitigate or accelerate climate change [1].

Northern peatlands are a specific ecosystem that plays an essential role in the global carbon cycle, which is disproportionally larger than their global coverage [4]. They are mostly found in the high latitudes of the northern hemisphere, where the temperature rises the fastest [1,5]. Carbon storage, which depends mainly on a low decomposition rate due to anoxic, low-pH conditions, is the most important function of peatlands. However, the assimilation of carbon by vegetation is not a negligible portion of the carbon cycle, as the final carbon balance is the difference between assimilation and release [4,6]. If the vegetation season starts earlier, the amount of assimilated carbon and the probability that the peatlands will remain the carbon sink increase [7,8].

The northern peatland vegetation comprises mainly mosses, graminoids, and ericoid shrubs [9]. Peat moss (*Sphagnum* spp.) and ericoid shrubs like bog cranberry (*Vaccinium oxycoccos* L.) can be characterized as evergreens, which means they start to be photosynthetically active as soon as the conditions in spring become favorable, without the need to grow leaf first [10,11]. Mosses and shrubs found in peatlands are constantly competing for the available light and nutrients. The recent warming and associated drying of peatlands led to the increased abundance of shrubs on account of mosses due to the evolutionary advantage they possess, such as roots or thick cuticula [10,12,13]. Climate change causes substantial changes in the peatland vegetation composition and its carbon assimilation [6]. Drying of peatlands alters the photosynthetic activity of their vegetation, with vascular plants being the main driver of the changes [14]. The production of peatland shrubs increases with higher temperatures and lower wetness, while moss production decreases with drying [15]. It has been shown that the increased temperature during the summer influences the photosynthetic potential of peatland vegetation negatively, with a stronger effect on peat moss than on bog cranberry [16]. However, it is unclear how different plant functional groups (PFGs) react to the increase in spring temperature and what mechanisms are behind the specific reactions of PFGs’ photosynthetic apparatus.

The physiological conditions of various components of photosystem II (PSII), elements of the electron transport chain, and the coordination between light-dependent and light-independent biochemical processes can be assessed through the analysis of fast chlorophyll fluorescence transient curves [17]. Although chlorophyll *a* fluorescence transient is mostly studied to reveal the negative impacts of different stressors on photosynthesis, we employed this technique to reveal the changes in the status of photosynthetic machinery in peatland plants during the spring warm-up [18]. In addition to naturally increasing temperature, we examined the effect of artificial increases in temperature in situ to simulate the future warmer climate. We performed this experiment aiming to clarify the impact of spring warming on the photosynthetic apparatus capacity of bog cranberry and peat moss because such knowledge is important for further refinement of peatland or global carbon cycle and vegetation models.

## 2. Results

### 2.1. Thermal Conditions of Experimental Plots

Due to cloudy conditions and high precipitation causing the water level to rise above the peatland’s surface, the W conditions had a considerable effect on the near-surface air temperature only from the last week of February onwards (Figure 1). Nigh-time heating mitigated the frost events when the minimal temperature of the coldest night in March (8 March—4 days before the first measurements) was 1.2 °C higher in W plots compared to C. The mean daily temperature one week before measurements was 1.2 °C, 0.9 °C, 0.8 °C, and 1.7 °C higher in W than C for the first, second, third, and fourth measurements, respectively. The cumulative difference in daily mean temperature between W and C rose from effectively 0 on 3 February to 21.6 °C the day before the first measurement and then to 29.3 °C, 44.0 °C, and 67.4 °C the day before the second, third, and fourth measurements, respectively.

### 2.2. Energy Partitioning at Photosystem II (PSII)

The significant differences in energy partitioning at PSII caused by warming were recorded only on the 13 March and the 3 May (Figure 2). While the actual quantum yield of photosystem II photochemistry (ϕPSII) was significantly increased by W conditions only on the 3 May for peat moss, the quantum yield of light-induced energy dissipation (ϕNPQ) was significantly higher for both PFGs on the 13 March and only for peat moss on the 3 May (Figure 2A,B). W induced a significantly lower quantum yield of light-independent energy dissipation (ϕNO) and significantly higher non-photochemical quenching of maximum fluorescence (NPQ) for both PFGs on the 13 March and the 3 May (Figure 2C,D).

ϕPSII of bog cranberry was increasing as the season progressed, with a 70% increase on the 12 April and doubling on the 3 May compared to the 13 March. Conversely, the decrease in ϕPSII to effective 0 on the 3 May was observed for peat moss. While bog cranberry used only 40% more of the trapped light for photochemistry than peat moss on the 13 March, this difference increased to 180% on the 12 April. The results from the 3 May could not be compared due to the almost 0 value of ϕPSII for peat moss (Figure 2A). While ϕNPQ of bog cranberry remained stable across all four measurements, it significantly dropped on the 3 May for peat moss. During the first three measurement days, peat moss dissipated around 50% more energy by ϕNPQ than bog cranberry (Figure 2B). ϕNO of bog cranberry was steadily decreasing with the progressing vegetation season when it was 20%, 38%, and 36% lower during the second, third, and fourth measurements, respectively, compared to the first measurement. Peat moss dissipated a remarkably stable fraction of the absorbed energy as ϕNO for the first three measurements. However, we recorded over a 3-fold increase in ϕNO for the measurement on the 3 May compared to the other measurements (Figure 2C). NPQ of bog cranberry increased by 64% between the first and second measurements and by 10 from the second to the third measurement, but it decreased by 11% from the third to the fourth measurement. At the same time, NPQ of peat moss increased only by 12% between the first two measurements and then decreased by 11% and by 84% between the second and third and the third and fourth measurements, respectively (Figure 2D).

### 2.3. The Fluorescence Transient Kinetics

The typical polyphasic fluorescence transient curves were recorded for bog cranberry in all measurement days and for peat moss in the first three measurements. However, the O-J-I-P curve of peat moss on the 3 May was altered when the change from J to I step was minimal, and the fluorescence intensity quickly dropped after reaching maximum, even to the values lower than minimal fluorescence (F_o_; Figure 3A). The double normalization of fluorescence transient by maximal fluorescence (F_m_) and F_o_ (calculated as (F_t_ − F_o_)/(F_m_ − F_o_)) revealed the differences, especially in the O-J step of the transient (Figure 3B). The further comparison of warming-induced differences in relative fluorescence for different steps of the O-J-I-P transient revealed the PFG-specific and date-dependent responses (Figure 4).

A positive K band occurring between 0.2 ms and 0.3 ms was observed for bog cranberry on the 3 May and for peat moss during all the measurements (Figure 4A,D). While W caused a positive change in the ratio of variable fluorescence of the O-J step (W_OJ_) for peat moss in all the measurement dates, the change was positive only during the first two measurements, followed by a negative change for the other two measurements in the case of bog cranberry. The apparent sudden changes in the difference of W_OJ_ observed between 0.6 ms and 1 ms are probably so prominent due to the large gap in the time series caused by the manufacturer’s equipment settings (Figure 4A,D). The negative difference with variable peak position and the highest prominence on the 13 March was observed for the W_JI_ of bog cranberry for all measurements. However, the warming-induced difference in W_JI_ for peat moss was negative during the March measurements, positive on the 12 April, and then sinusoid on the 3 May, with the first negative trend between 3.1 ms and 9.3 ms and followed by a positive trend until 30 ms (Figure 4B,E). The difference in W_IP_ (ΔW_IP_) caused by W was characterized by a positive trend with a progressive peak shift to a later time and an increasing magnitude of the difference with the season advancement. Negative change in ΔW_IP_ for peat moss was found in March, while positive change was observed on the 12 April and the 3 May (Figure 4C,E).

Several parameters assessing the electron transport and the status of the reaction centers (RCs) were calculated from the fast chlorophyll *a* transient to facilitate statistical analysis. The general trend for bog cranberry suggests decreasing or diminishing warming-induced differences with progressing vegetation season (Figure 3 and Figure 5). Relative variable fluorescence at step J, relative variable fluorescence at step I, normalized area above O-J curve, energy flux trapped by one active RC at time 0 (TRo/RC), rate of electron transport by one active RC at time 0 (ETo/RC), efficiency of electron transport beyond plastoquinone (Ψo), and approximated initial slope of the fluorescence transient (Mo) of bog cranberry were not significantly altered by increased temperature in any of the measurement days. The significantly higher maximum fluorescence normalized by minimum fluorescence (F_m_/F_o_), causing also higher maximum efficiency of the water diffusion reaction on the donor side of photosystem II (F_v_/F_o_), persisted throughout the observed period. Nevertheless, the difference was decreasing from 47% on the 13 March to 16% on the 3 May for F_m_/F_o_ and from 86% to 25% for F_v_/F_o_ (Figure 5A–D). Additionally, W significantly increased the maximum quantum yield of photosystem II photochemistry (ϕPo) and photosystem II performance index on an absorption basis (PI ABS) while decreasing the normalized area above the O-J-I-P curve, number of plastoquinone reductions from time 0 to time of reaching maximum fluorescence (N), absorption flux per one active RC at time 0 (ABS/RC), energy flux not intercepted by reaction center at time 0 (DIo/RC), and quantum yield of energy dissipation at time 0 (ϕDo) of bog cranberry during the March measurements (Figure 3A,B). ϕPo and PI ABS were significantly increased, and ABS/RC, DIo/RC, and ϕDo were decreased by higher temperature also on the 12 April (Figure 5C). On the 3 May, only the normalized area above the O-J-I-P curve and N were significantly lower in leaves of bog cranberries growing in W plots compared to C (Figure 5D).

Unlike for bog cranberry, there was no chlorophyll *a* fluorescence transient-derived parameter that was significantly altered by W in all campaigns. W induced similar changes in the photosynthetic apparatus of peat moss on the 13 March as it did for bog cranberry when F_m_/F_o_, F_v_/F_m_, ETo/RC, ϕPo, Ψo, and PI ABS increased, and relative variable fluorescence at step J, the normalized area above the O-J-I-P curve, N, ABS/RC, DIo/RC, and ϕDo decreased as a result of higher temperature (Figure 5E). Only a decrease in the normalized areas above the O-J and the O-J-I-P curves and N and an increase in TRo/RC for peat moss growing under W conditions were significant compared to C on the 22 March (Figure 5F). W caused again more significant alterations in peat moss photosynthetic apparatus in April and May. Relative variable fluorescence at step J, F_m_/F_o_, F_v_/F_o_, ABS/RC, ETo/RC, DIo/RC, ϕPo, Ψo, and ϕDo were significantly changed by increased temperature in both of the last two measurement days, while PI ABS of moss from W plots was significantly higher only on the 12 April, and relative variable fluorescence at step I, the normalized area above the O-J-I-P curve, and N were significantly lower only on the 3 May (Figure 5G,H).

## 3. Discussion

The beginning of 2024 was unusual in terms of temperature, with every month from January to May being the warmest on record [3]. The 9 February was the last day when the average daily temperature in the studied peatland was below 0 °C (Figure 1A). This could be marked as the beginning of the vegetation season for the evergreen plants in 2024.

The low temperature and high light intensity during winter and early spring could damage the photosynthetic apparatus of evergreen plants. Therefore, the decreasing photosynthetic efficiency and structural changes on PSII protect the plants from damage to photosystems in winter. This decrease is a reversible change, and the photosynthetic efficiency is restored in the spring when temperature rises [18,19,20,21]. The increasing ϕPSII of bog cranberry from the 22 March to the 12 April confirms that the season of photosynthetic activity for this shrub started in the late March to early April period. However, the remarkable stability of peat moss ϕPSII shows that its season started already before the 13 March. That may be because the peat moss is photosynthetically active when the temperature is above 0 °C, and it continues the growth over the wintertime, which suggests no permanent downregulation of PSII as in the case of vascular plants [22,23]. Indeed, we found that for the first three measurements gathered during the March–April period, peat mosses dissipated the majority of the excess energy by light-dependent heat dissipation connected with the changes in pH gradient and the xanthophyll cycle (Figure 2). This indicates that peat mosses are able to quickly react on even shorter periods of favorable conditions during winter by increased rates of photochemistry [24]. Therefore, peat mosses are a major contributor to the peatland vegetation carbon uptake during the beginning of vegetation season, before ϕNO of shrubs relaxes due to long enough periods of temperatures above the threshold [25]. Nevertheless, the situation changed during the measurements on the 3 May, when we observed a drop of ϕPSII to almost 0 and a dramatic increase in ϕNO, suggesting that the peat moss switched from photosynthetic activity to sustained energy dissipation (Figure 2). Conversely, the low ϕPSII of bog cranberry was maintained in March by higher ϕNO, which decreased with time and also by increased temperature during the first and last measurements, while the ϕPSII was increasing. At the same time, ϕNPQ remained almost constant. These changes in the absorbed energy partitioning on PSII show that bog cranberries rely on more sustained, light-independent energy dissipation upregulation during cold months, which provides more constant protection of photosynthetic machinery that relaxes slower compared to the strategy employed by peat moss [20,26]. Our findings are seemingly in dispute with the previous research suggesting that the shrubs of the *Vaccinium* genus maintain photosynthetic activity during the winter without downregulation [19,27]. However, those studies were performed on shrubs overwintering under the snow cover that protects them from excessive light or provides insulation from frost. On the contrary, the peatland site used for this study is snow-free for most of the winter, and the plants are exposed to adverse environmental conditions.

Although higher values of NPQ are generally connected to stress conditions, this is not the case in our study, as the NPQ does not increase as a result of increased ϕNPQ but due to decreasing ϕNO [28]. Interestingly, when the peat moss seems to become stressed on the 3 May, the NPQ value decreases from above 3 to below 1 (Figure 2D). Experimental W induced changes in the energy dissipation pathways during the first and last measurements. Further analysis revealed that the days when significant differences in energy partitioning occurred were the days preceded by a week when W plots were, on average, more than 1 °C warmer compared to C. It seems that the lower difference was not enough to influence the major energy distribution processes, but, as discussed below, several other processes were significantly impacted.

The fluorescence transient curve of evergreens undergoes changes in shape during the spring warm-up. The curve has a low magnitude and misses a clear I step in winter, and it achieves its typical O-J-I-P shape as the temperature rises and the physiological activity of plants starts [21]. As our measurements started at the beginning of the meteorological spring, we observed the typical O-J-I-P shape of the curve from the beginning of our measurements for both PFGs. However, the transient curve of C bog cranberries in March was rather flat and became its typical contours only in April. Moreover, the 1.5-fold bigger magnitude of the W plots bog cranberry’s curve in March compared to the C plants during the same measurements that was reached by C plants only in April suggests the earlier onset of the photosynthetic activity in W plots (Figure 3A). Similarly to previous work [16], we found that the chlorophyll *a* fluorescence transient of peat moss rises less prominently than the fluorescence of the peatland vascular plants, particularly the bog cranberry (Figure 3A). The fast rise of the fluorescence from the O to J step for peat moss on the 3 May resembles the O-J-I-P curves of plants treated by the electron transport inhibitors. The transient curves of mosses grown in both C and W conditions point to disrupted electron flow beyond plastoquinone’s Q_A_ binding site [29]. The appearance of the positive K band in peat moss and the negative K band in bog cranberry in May points to the temperature-influenced changes in the electron transport and oxygen-evolving complex (Figure 4) [30]. However, the absence of the K step in the fluorescence transient curve indicates no severe stress on the plants [31]. On the contrary, the significantly higher F_v_/F_o_ for plants under W conditions is indicative of increased activity of the oxygen-evolving complex (Figure 5). The warming-induced changes in the oxygen evolution are also visible from the differences in the W_JI_ observer for both PFGs (Figure 4B) [32].

W significantly decreased the functional size of the PSII antenna in bog cranberries, as suggested by ABS/RC [33]. The difference was the most pronounced in the early spring and then decreased and even disappeared as the season progressed. This could be because the lower air temperature of the C plots in early spring did not allow for all the RCs of PSII to be active. However, with increasing temperature, as the season progressed, it became warm enough to activate all the RCs, and the photosynthetic apparatus of cranberries reached very similar status for C and W in May (Figure 5D). The increased ϕPo of bog cranberry in W plots during the March and April measurements was due to the faster conversion of PSII RCs from the heat sinks back to the active state. This is apparent as Mo, representing the net rate of the RCs’ closure, TRo/RC or ETo/RC were not altered, but the DIo/RC of W plants decreased significantly, which indicates lower non-photochemical energy dissipation [34,35]. The progressive and temperature-dependent increase in ϕPo found in this work is in concordance with the previous study of evergreen plants by others [21]. Similarly to bog cranberry, W decreased the size of the peat moss PSII antenna and increased the ϕPo and DIo/RC. However, it also caused changes in the electron transport rate, as suggested by relative variable fluorescence at step J, ETo/RC, and Ψo [28]. The peat moss exhibited more complex behavior than the bog cranberry, as there was not one trend of decreasing differences between W and C during the spring. In fact, the differences between temperature regimes disappeared on the 22 March but started to become larger again in April and May (Figure 5E–H).

The uniform relative variable fluorescence at step J values between W and C bog cranberries indicated no change in the plastoquinone pool size, which means that the electrons could be transferred to the dark reaction sites without obstructions in both temperature regimes [33,36]. However, despite the single plastoquinone’s Q_A_ binding site reduction rate being unchanged by W as visible from the normalized area above the O-J curve, the number of turnovers (as suggested by the normalized area above the O-J-I-P curve and N) per single RC decreased due to the abovementioned increase in the number of active PSII RCs [28]. The absence of differences in relative variable fluorescence at step I suggests that the pool size of photosystem I acceptors remained stable irrespective of treatment or date for bog cranberry on all days and the peat moss during the first three measurements. However, the increase in relative variable fluorescence at step I for peat moss C compared to the other values of relative variable fluorescence at step I recorded for peat moss could be interpreted as an increased photosystem I acceptor pool. Despite the size of the photosystem I acceptor pool not changing due to the W, the reduction rate of the photosystem I end electron acceptors was altered, as indicated by the difference in the W_IP_ dynamics (Figure 4C,F) [29,34]. The reduction rate of bog cranberries photosystem I acceptors was increased by W in all of the measurement days, while W enhanced the photosystem I reduction rate in peat moss during the April and May measurements but decreased it negligibly in March (Figure 4C,F).

The warming-induced increase in PI ABS of bog cranberry reflects mainly the decreasing size of the PSII antenna, as the electron trapping efficiency and their transport beyond plastoquinone’s Q_A_ binding site were not significantly altered. However, PI ABS of peat moss was increased by W due to the synergistic effect of antenna size decrease and improved electron trapping and transport [37]. It is not surprising that PI ABS was the parameter with the most pronounced warming-induced changes, as it is the most sensitive parameter representing the overall status of the photosynthetic apparatus, which is considerably temperature-dependent [17,21,30]. Despite the visible relative difference in PI ABS of moss on the 3 May, the difference was insignificant as the values of the index approached 0. That underscores the apparent poor status of peat moss photosynthetic apparatus during the last measurements (Figure 5H). The unusual values of chlorophyll *a* fluorescence recorded on the 3 May for peat moss could be explained by the higher temperature, which caused drying of the moss surface during the dark adaptation in the clip; therefore, causing the stress similar to the application of the electron transport inhibitors. However, this theory needs to be verified by future, well-designed experiments. Should it be true, the specific clips for mosses need to be designed to obtain reliable data.

## 4. Material and Methods

### 4.1. Study Site

The data were collected at the experimental station localized in Rzecin peatland (52°45′41″ N, 16°18′35″ E, 54 m a.s.l.), western Poland, where the warming and reduced precipitation experiment was established in 2017 (Figure 6). A detailed description of the site is provided by [38,39]. The plots of control (C) and warming (W) conditions of the CR (dominant graminoid is beaked sedge; *Carex rostra* Stokes) site were used for this study (Figure 6). The vegetation in the experimental plots is natural and consists of bog cranberry, peat mosses, beaked sedge, and swamp horsetail (*Equisetum fluviatile* L.), with only cranberries and mosses being of evergreen nature. The peat moss layer is dominated by the species *Sphagnum angustifolium* (Warnst.) C.E.O. Jensen, *S. fallax* (Klinggr.) Klinggr., and *S. teres* (Schimp.) Ångstr. The species are present in the plots in variable ratios representing the natural variability of the vegetation and have been present on site for at least 3 decades [40,41].

C plots are exposed to ambient conditions, while the increased temperature of W plots is reached passively by open-top chambers (OTCs) during the daytime and actively by a 100 W infrared heater during the night. All the plots are randomly distributed within the experimental site in triplicate. The air temperature was measured next to every plot 30 cm above the surface by HygroVue5 thermohygrometers (Campbell Sci., Logan, UT, USA) and recorded every half an hour on a datalogger CR1000 (Campbell Sci., Logan, UT, USA) [38,39].

Due to the higher-than-usual precipitation of 381 mm in the period between 1 September 2023 and 29 February 2024, the hollows of the site were underwater, and around 70% of the surface was above water level.

### 4.2. Chlorophyll a Fluorescence Measurements

Fast chlorophyll *a* fluorescence kinetics and the pulse–amplitude modulated fluorescence measurements were performed by FluorPen FP 110/D (Photon System Instruments, Drasov, Czech Republic) with detachable dark adaptation clips. The measurements were performed 4 times during the meteorological spring of 2024: on the 13 and 22 March, the 12 April, and the 3 May under cloudy conditions. The fluorescence of bog cranberry was measured on the adaxial side of the leaves of different plants in the plot. Measurements of peat moss fluorescence were derived from the freshly detached capitula’s top surface. As the peat moss species occurring in the plots are closely related, they were treated as a genus, *Sphagnum* spp., without further classification into species. After 25 min of dark adaptation, 5 samples of each PFG in the plot were measured by the OJIP protocol, and 5 samples were analyzed by the NPQ3 protocol provided by the manufacturer. The OJIP protocol is a 2-s-lasting fast chlorophyll *a* fluorescence kinetics measurement, while NPQ3 is pulse–amplitude modulated fluorescence measurement under artificial actinic light that takes 200 s and comprises 10 pulses. The measurement light was set to 85%, while the actinic light for the NPQ3 protocol was 300 μmol m^−2^ s^−1^. The wavelength of the used light was 455 nm. Parameters calculated from the measured fluorescence values are summarized in Table 1.

### 4.3. Statistical Analysis

The data were visually screened for incorrect measurements in Microsoft Excel (Microsoft Corporation, Redmond, WA, USA). The measurements with stable low fluorescence, indicating measurements of closed darkening clips, were removed before the statistical analysis. The significance of differences between C and W conditions was analyzed by Student’s *t*-test for every PFG and date separately. All the measurements of the same PFG on the same day from all 3 plots were analyzed together. The analysis was carried out in RStudio version 2023.12.1+402 (RStudio Inc., Boston, MA, USA). As many of the cranberry leaves are smaller than the area of the darkening clip opening, only values normalized by F_o_ were analyzed and presented in the publication.

## 5. Conclusions

The unusually warm spring of 2024 shifted the vegetation phenology of plants. However, the experimental warming of the peatland ecosystem in situ proved that further shifts are possible in the near future for peat moss and bog cranberry. We showed that different PFGs (mosses and shrubs) rely on different strategies of photosynthetic apparatus protection during winter, which is demonstrated by higher ϕNO and lower ϕNPQ of bog cranberries compared to peat moss in very early spring. This indicates that peat mosses are able to quickly react on even shorter periods of favorable conditions during winter by increased rates of photochemistry, while bog cranberries utilize the strategy of more constant protection of photosynthetic machinery that relaxes comparatively slower. The switch from the inactive winter status to spring photosynthetic activity is faster if the air temperature is higher. The process of photosynthesis activation in bog cranberries is accompanied by the activation of RCs, decreased heat dissipation, and increased overall efficiency of PSII, all of which were found to be temperature-dependent.

The results of this study help to understand the observed shrub encroachment of peatlands with elevated temperatures. A stronger positive effect of warming on bog cranberry than on peat moss during the spring found in this study and a stronger negative effect of summer warming reported previously show that the photosynthetic apparatus of cranberries is much better equipped for the future with higher temperatures.

Our findings contribute to the understanding of the morphological and physiological advantages of the ericaceous shrubs over the peat moss that allow them to take over bigger and bigger portions of the peatlands as the world warms up. The future vegetation and carbon models should take this information into account to accurately predict the peatland functions under a changing climate.

## Figures and Tables

**Figure 1 plants-13-03246-f001:**
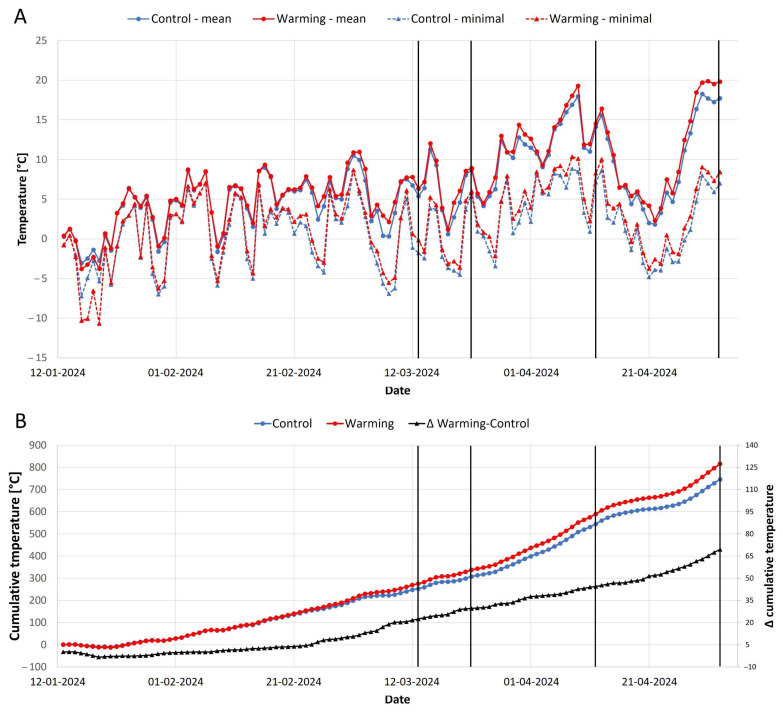
Mean and minimal daily air temperature (**A**), cumulative mean daily air temperature, and the difference (Δ) in cumulative temperatures (**B**) of warming and control plots measured 30 cm above the peatland surface in the period of 2 months before the first measurements until the last measurements. The thick black horizontal lines indicate the days of measurements (from left to right: 13 March, 22 March, 12 April, 3 May).

**Figure 2 plants-13-03246-f002:**
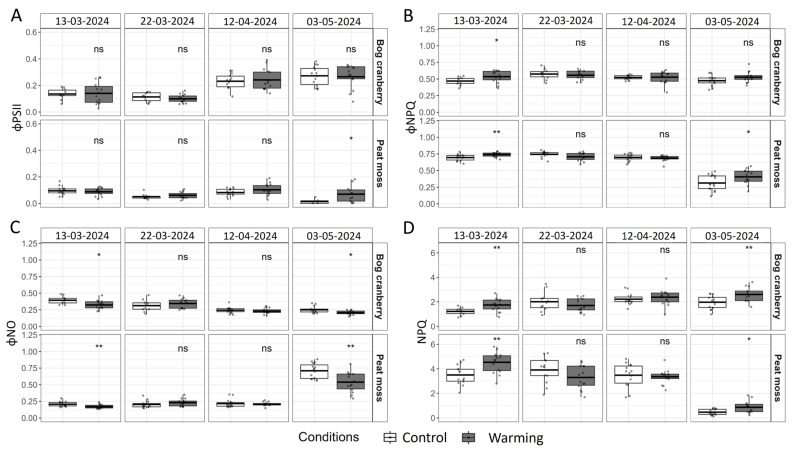
The actual quantum yield of photosystem II photochemistry (ϕPSII; (**A**)), the quantum yield of light-induced energy dissipation (ϕNPQ; (**B**)), the quantum yield of light-independent energy dissipation (ϕNO, (**C**)), and non-photochemical quenching of maximum fluorescence (NPQ; (**D**)) measured at 300 μmol m^−2^ s^−1^ from bog cranberry and peat moss subjected to control and warming conditions. The thick line in the boxplot represents the mean, while the points represent single measurements. The signs denote the significance of the difference between control and warming for each plant functional group and date separately, where “ns” is non-significant, * is <0.05, ** is <0.01.

**Figure 3 plants-13-03246-f003:**
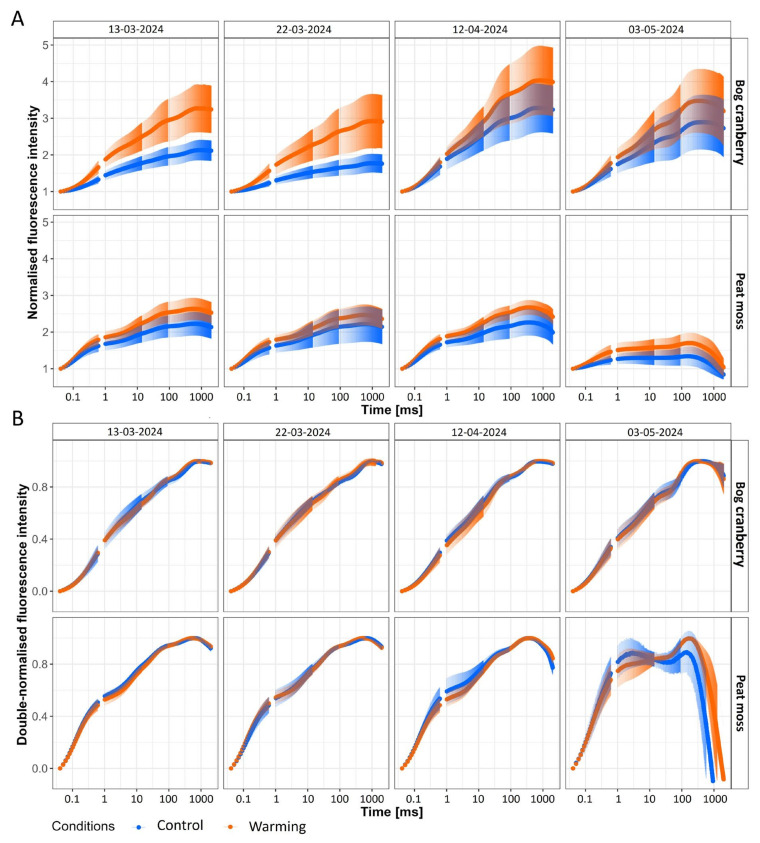
Fast chlorophyll *a* fluorescence transient curves normalized by minimal fluorescence (F_t_/F_o_; (**A**)) and double normalized by minimal and maximal fluorescence (F_t_ − F_o_)/(F_m_ − F_o_; (**B**)) for bog cranberry and peat moss (in rows) growing under control and warming conditions measured during different days (in columns). The points represent the mean, and the bars represent the standard deviation, *n* = 15.

**Figure 4 plants-13-03246-f004:**
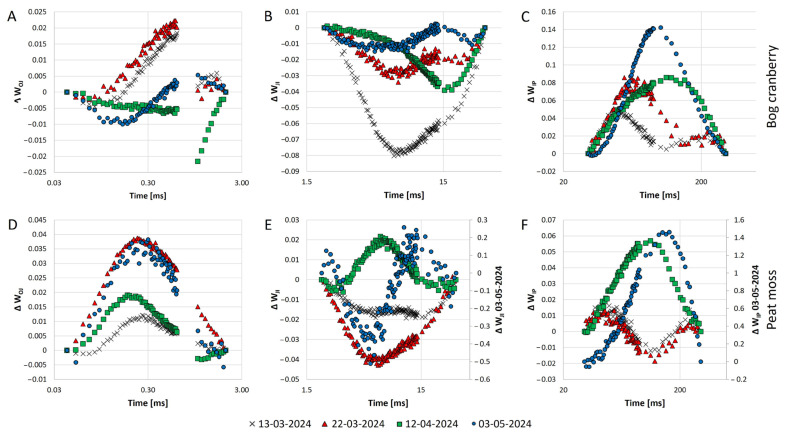
The warming-induced change (calculated as variable fluorescence (V_t_) warming—V_t_ control) in the ratio of variable fluorescence to the amplitude of F_J_ − F_o_ (ΔW_OJ_; (**A**)), F_I_ − F_J_ (ΔW_JI_; (**B**)), and F_P_ − F_I_ (ΔW_IP_; (**C**)) for bog cranberry, and the ratio of V_t_ to the amplitude of F_J_ − F_o_ (ΔW_OJ_; (**D**)), F_I_ − F_J_ (ΔW_JI_; (**E**)), and F_P_ − F_I_ (ΔW_IP_; (**F**)) for peat moss. Note that there is a secondary axis at panels E and F for 3 May 2024 due to the different magnitudes of the change. The points represent the mean of all measurements for a given plant functional group and day, *n* = 15.

**Figure 5 plants-13-03246-f005:**
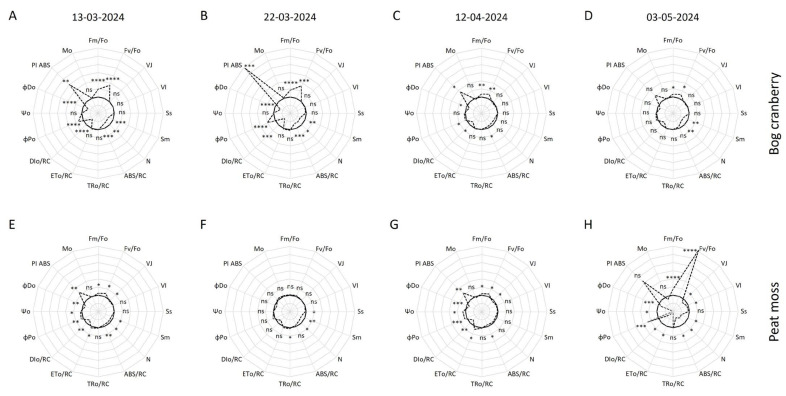
Spider plots presenting the parameters calculated from the chlorophyll *a* fluorescence transient normalized by the respective control for bog cranberry (**A**–**D**) and peat moss (**E**–**H**) on 13 March (**A**,**E**), 22 March (**B**,**F**), 12 April (**C**,**G**), and 3 May (**D**,**H**); the full line indicates the control, while the dashed line indicates warming. The signs denote the significance of the difference between control and warming for each plant functional group and date separately, where “ns” is non-significant, * is <0.05, ** is <0.01, *** is < 0.001, and **** is <0.0001. The distance between concentric circles is 0.5, while the control, thick full line is at value 1. The displayed parameters are: maximum fluorescence normalized by minimum fluorescence (F_m_/F_o_), maximum efficiency of the water diffusion reaction on the donor side of photosystem II (F_v_/F_o_), relative variable fluorescence at the step J (V_J_), relative variable fluorescence at step I (V_I_), the normalized area above O-J curve (Ss), the normalized area above the O-J-I-P curve (Sm), number of plastoquinone reductions from time 0 to time of reaching maximum fluorescence (N), absorption flux per one active reaction center at time 0 (ABS/RC), energy flux trapped by one active reaction center at time 0 (TRo/RC), rate of electron transport by one active reaction center at time 0 (ETo/RC), energy flux not intercepted by reaction center at time 0 (DIo/RC), the maximum quantum yield of photosystem II photochemistry (ϕPo), efficiency of electron transport beyond plastoquinone (Ψo), quantum yield of energy dissipation at time 0 (ϕDo), photosystem II performance index on an absorption basis (PI ABS), and approximated initial slope of the fluorescence transient (Mo).

**Figure 6 plants-13-03246-f006:**
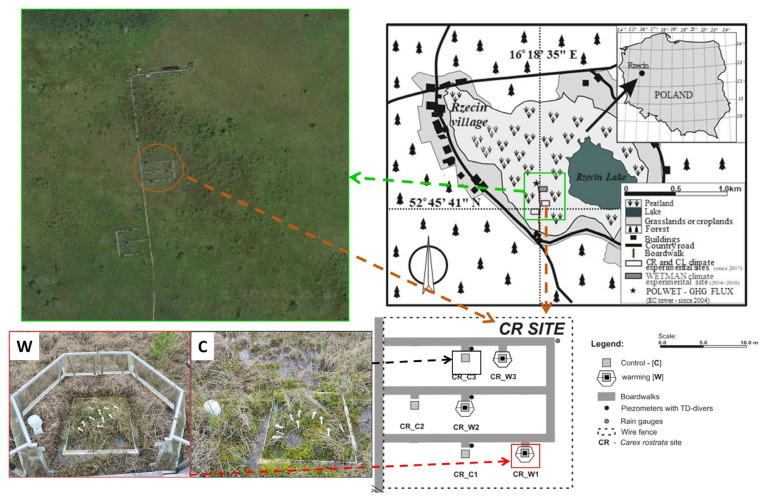
The location of Rzecin peatland in Poland and the location and experimental design of the CR site and its plots used for this study. Panels denoted W and C present representative warming and control plots, respectively, during the dark adaptation of bog cranberry leaves. Please note that the infrared heater was removed from the warming plot during the measurement as it is working only during the night-time (part of the figure was adapted from Górecki et al. [38], CC BY-NC-ND license).

**Table 1 plants-13-03246-t001:** Summary of chlorophyll *a* fluorescence-derived parameters with their abbreviations and the measurement protocols used for their calculation. The nomenclature follows Kalaji et al. (2017) and Strasser et al. (2000) [28,35].

Abbreviation	Name of Parameter	Measurement Protocol
F_o_	Minimal fluorescence	OJIP and NPQ3
F_m_	Maximal fluorescence	OJIP and NPQ3
ϕPo	Maximum quantum yield of photosystem II photochemistry	OJIP and NPQ3
ϕPSII	Actual quantum yield of photosystem II photochemistry (at 300 μmol m^−2^ s^−1^)	NPQ3
ϕNPQ	Quantum yield of light-induced energy dissipation (at 300 μmol m^−2^ s^−1^)	NPQ3
ϕNO	Quantum yield of light-independent energy dissipation (at 300 μmol m^−2^ s^−1^)	NPQ3
NPQ	Non-photochemical quenching of maximum fluorescence (at 300 μmol m^−2^ s^−1^)	NPQ3
ϕDo	Quantum yield of energy dissipation at time 0	OJIP
F_m_/F_o_	Maximum fluorescence normalized by minimum fluorescence	OJIP
F_v_/F_o_	Maximum efficiency of the water diffusion reaction on the donor side of photosystem II	OJIP
V_J_	Relative variable fluorescence at step J	OJIP
V_I_	Relative variable fluorescence at step I	OJIP
Ψo	Efficiency of electron transport beyond plastoquinone	OJIP
ABS/RC	Absorption flux per one active reaction center at time 0	OJIP
TRo/RC	Energy flux trapped by one active reaction center at time 0	OJIP
ETo/RC	Rate of electron transport by one active reaction center at time 0	OJIP
DIo/RC	Energy flux not intercepted by reaction center at time 0	OJIP
Sm	Normalized area above the O-J-I-P curve	OJIP
Ss	Normalized area above O-J curve	OJIP
Mo	Approximated initial slope of the fluorescence transient	OJIP
N	Number of plastoquinone reductions from time 0 to time of reaching maximum fluorescence	OJIP
PI ABS	Photosystem II performance index on an absorption basis	OJIP

## Data Availability

The data presented in this study are available on request from the corresponding author.

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
