# Peer review of "Photosynthetic Responses of Peat Moss (*Sphagnum* spp.) and Bog Cranberry (*Vaccinium oxycoccos* L.) to Spring Warming"

_plants, 2024, doi:10.3390/plants13223246_

Round 1
Reviewer 1 Report
Comments and Suggestions for Authors
In this study, Michal Antala and colleagues examined the impact of experimental warming on the status of photosynthetic apparatus of peat moss (Sphagnum spp.) and bog cranberry (Vaccinium oxycoccos L.) during the early vegetation season and studied the differences in the winter to early spring transition of peat moss and bog cranberry photosynthetic activity. This manuscript is interesting and I have the following comments:
1, For the title, the present version looks like a review, and I suggest to use “Analysis on Photosynthetic Responses of Peat Moss (Sphagnum spp.) and Bog Cranberry (Vaccinium oxycoccos L.) on Peatland to Spring Warming”.
2, For the abstract, speculative sentences like “However, bog cranberry needs a period of warmer temperature to reach full activity due to the sustained, non-regulated, heat dissipation during winter. Therefore, bog cranberry can benefit more from early spring warming, as its activity is sped up more compared to peat moss. This will probably result in faster shrub encroachment of the peatlands in the warmer future.” should be rephrased, and more specific data should be included.
3, For the introduction, more studies on the climate change and peatland vegetation should be introduced.
4, For the results, all data are obtained from the year 2024, what is the situation for 2021, 2022, and 2023. I strongly to include these years data as controls.
5, In addition, too many abbreviations like were employed in the present version, which hinders my understanding of this study. Authors should consider to rephrase the text to solve this concern.
Author Response
Comment 1: For the title, the present version looks like a review, and I suggest to use “Analysis on Photosynthetic Responses of Peat Moss (Sphagnum spp.) and Bog Cranberry (Vaccinium oxycoccos L.) on Peatland to Spring Warming”.
Response 1: Thank you for your suggestion. We agree that the first part of the title was very general and made it look like a review. Therefore, we removed the part before the column, leaving only the more exact part to deliver the point of the article more clearly.
Comment 2: For the abstract, speculative sentences like “However, bog cranberry needs a period of warmer temperature to reach full activity due to the sustained, non-regulated, heat dissipation during winter. Therefore, bog cranberry can benefit more from early spring warming, as its activity is sped up more compared to peat moss. This will probably result in faster shrub encroachment of the peatlands in the warmer future.” should be rephrased, and more specific data should be included.
Response 2: In the revised version, we supported the statements with more specific data from the article and softened the derived conclusions that are not directly supported by results.
Comment 3: For the introduction, more studies on the climate change and peatland vegetation should be introduced.
Response 3: We included more information about the impacts of climate change on peatland vegetation with new references to the introduction.
Comment 4: For the results, all data are obtained from the year 2024, what is the situation for 2021, 2022, and 2023. I strongly to include these years data as controls.
Response 4: Thank you for this suggestion. We agree that the inclusion of data for more years would strengthen the conclusions of our study. Unfortunately, we did not measure the presented parameters in the previous years. In our opinion, our findings are significant and unique as we are not aware of any similar study published to this date. The spring of 2024 was special as it was the warmest period ever recorded. Moreover, our manipulation increased the temperature and we were able to show that future warming will further impact the peatland vegetation. Therefore, we would like to ask for consideration of the publication only with the presented data for the year 2024.
Comment 5: In addition, too many abbreviations like were employed in the present version, which hinders my understanding of this study. Authors should consider to rephrase the text to solve this concern.
Response 5: Thank you for pointing out the issue which lowered the clarity of the previous version of the manuscript. In the revised manuscript, we used the full names of some of the parameters and the parts of the photosynthetic apparatus to decrease the number of abbreviations used in the text. We admit that the end of the results section still contains a significant amount of abbreviations; however these are standard abbreviations used in chlorophyll a fluorescence transient studies, and all their full names are introduced in the mentioned section and also presented in captions of figure 5. The use of the full names of all the parameters every time would make this section unnecessarily long; therefore, we think that the current use of abbreviations in the revised version is justified. We hope that the decreased used of abbreviations made the discussion clear and the readers will understand the interpretation of our results.
Reviewer 2 Report
Comments and Suggestions for Authors
The manuscript title “Climate Change and Peatland Vegetation: Photosynthetic Responses of Peat Moss (Sphagnum spp.) and Bog Cranberry (Vaccinium oxycoccos L.) to Spring Warming” is conducted well and has scientific worth. Increasing global temperatures due to climate change causes significant impacts on the photosynthesis of peatland vegetation; this study focuses on how this climate change affects the photosynthesis of peatland vegetation.
Summary of Manuscript:
This study focuses on two common plant functional groups in peatlands: peat moss (Sphagnum spp.) and bog cranberry (Vaccinium oxycoccos L.). We investigated the effects of experimental warming on their photosynthetic apparatus during the early vegetation season and examined the differences in photosynthetic activity from winter to early spring. Peat moss initiates photosynthetic activity earlier in the season, relying on light-dependent energy dissipation throughout winter. In contrast, bog cranberry requires a period of warmer temperatures to achieve full photosynthetic activity due to sustained, non-regulated heat dissipation during winter. Bog cranberry benefits more from early spring warming, leading to a more significant increase in activity compared to peat moss. This differential response may accelerate shrub encroachment in peatlands as temperatures rise.
The findings suggest that vegetation and carbon models should incorporate these results to better predict peatland functions under changing climate conditions, particularly regarding the interactions between plant species and their responses to warming.
I have few suggestions to improve the current version of the manuscript.
Comments for authors are as follows:
1- Line 365: “A detailed description of the site is” some necessary details/ description should be added here, related to this experiment.
2- Line 367- 370: “The vegetation in the experimental plots consists of bog cranberry, peat mosses, beaked sedge, and swamp horsetail (Equisetum fluviatile L.), with only cranberries and mosses being of ever-green nature. The peat moss layer is dominated by the species Sphagnum angustifolium (Warnst.) C.E.O. Jensen, S. fallax (Klinggr.) Klinggr., and S. teres (Schimp.) Ångstr.” Please provide more information about the how many pots or how many bog cranberries, peat mosses, beaked sedge, and swamp horsetail etc., in one experimental plot?
3- How old were the “bog cranberry, peat mosses, beaked sedge, and swamp horsetail etc.,”?
4- Line 372: “ambient conditions” please provide some details of ambient conditions….
5- Line 375: “The air temperature was measured next to every plot” the temperature belongs to method section; I suggest adding the temperature information in the methods.
6- I suggest the authors split the conclusions section into ‘conclusion’ and ‘future prospects.
Author Response
Comment 1: Line 365: “A detailed description of the site is” some necessary details/ description should be added here, related to this experiment.
Response 1: In the revised version, we provided more information about the site and the conditions. Moreover, we added a figure that clarifies the experimental design and better illustrates the vegetation of the experimental site.
Comment 2: Line 367- 370: “The vegetation in the experimental plots consists of bog cranberry, peat mosses, beaked sedge, and swamp horsetail (Equisetum fluviatile L.), with only cranberries and mosses being of ever-green nature. The peat moss layer is dominated by the species Sphagnum angustifolium (Warnst.) C.E.O. Jensen, S. fallax (Klinggr.) Klinggr., and S. teres (Schimp.) Ångstr.” Please provide more information about the how many pots or how many bog cranberries, peat mosses, beaked sedge, and swamp horsetail etc., in one experimental plot?
Response 2: The experimental plots contain only natural vegetation. There are no pots or planted plants. We added clarification to the site description and added Figure 6, which also depicts representative experimental plots. The ratios of the present plants somewhat vary among the plots to represent the natural variability of the given vegetation. This information was also added to the revised manuscript.
Comment 3: How old were the “bog cranberry, peat mosses, beaked sedge, and swamp horsetail etc.,”?
Response 3: It is practically impossible to determine the age of the present plants as the measured species are perennial, and they have annual increments of new biomass from the old stems that get buried in peat. However, based on the palaeobotanical research done by our colleagues, these species have been present in their present location for at least the last 3 decades. We added this information and included the references to the revised manuscript.
Comment 4: Line 372: “ambient conditions” please provide some details of ambient conditions….
Response 4: The thermal conditions of the plots are described in detail in the first section of the results. We added the hydrological description to the methodology of the revised manuscript.
Comment 5: Line 375: “The air temperature was measured next to every plot” the temperature belongs to method section; I suggest adding the temperature information in the methods.
Response 5: We decided to present the thermal conditions at the beginning of the results for two reasons: a) temperature is the main factor examined in this study, and because the methodology section is coming at the end of the article, the reader should get the information about the temperature difference between control and warming before the other results are presented; b) as the experiment is conducted in the natural conditions where the warming cannot be exactly set to given temperature or temperature difference, the air temperature data is a result of the manipulation rather than a simple conditions description.
Comment 6: I suggest the authors split the conclusions section into ‘conclusion’ and ‘future prospects.
Response 6: Thank you for the suggestion. As we understood from the journal template that it is not usual to make a separate section for Future perspectives, we separated the last paragraph of the Conclusions section to deliver the future perspectives more clearly.
Round 2
Reviewer 1 Report
Comments and Suggestions for Authors
Authors have answered my questions in the revision.